# Computational Sedimentation Modelling Calibration: a tool to measure the settling velocity at different gravity conditions

Nikolaus J. Kuhn[1], Federica Trudu[1,2]

[1]Physical Geography and Environmental Change, Department of Environmental Sciences, University of Basel, Klingerbergstrasse 27, 4056 Basel, Switzerland

[2] University of Applied Science of Southern Switzerland-SUPSI, Department of Innovative Technologies-DTI, Via la Santa 1, 6962 Viganello, Switzerland

*Correspondence to*: Nikolaus J. Kuhn (nikolaus.kuhn@unibas.ch)

**Abstract**. Research in zero or reduced gravity is essential to prepare and support planetary sciences and space exploration. In this study, an instrument specifically designed to measure the settling velocity of sediment particles under normal, hyper-, and reduced gravity conditions is presented. The lower gravity on Mars potentially reduces drag on particles settling in water, which in turn may affect the texture of sedimentary rocks forming in a standing or moving body of water with settling particles. An environment to test such potential errors are parabolic flights which offer reduced gravity for up to 30 seconds. Exact tracing of particle tracks while settling is essential to assess the impact of gravity on flow hydraulics, drag and settling velocity. In this study, we present an advanced version of previous instruments, including the approach to particle tracking and track analysis. The trajectories of particles settling in water were recorded under reduced Martian and lunar gravity, the hypergravity phases during the pull-up of the plane and at terrestrial gravity on Earth. The data were used to compute the terminal settling velocity of isolated and small groups of particles and compared to the results calculated using a semi theoretical formula derived in 2004 by Ferguson and Church (Ferguson & Church, 2004). The analysis showed that with improved design of settling chambers, particle recording and tracking, a highly precise measurement of settling velocity is possible. This illustrates that the parabolic flight environment is not just suited for broad, qualitative comparisons between gravity environments, but also highly precise data acquisition on flow hydraulics associated with particle settling.

## 1 Introduction

Conducting research in zero or reduced gravity helps simulate the conditions experienced in outer space and other planetary bodies, such as the Moon and Mars. This is crucial for understanding how various phenomena, materials, and biological processes change under the influence of gravitational accelerations different from Earth's. For example, with reduced gravitational force, fluid flow dynamics and other physical quantities relevant to the morphology of planets and moons in our solar system, change. One of these physical quantities is the settling velocity of solid particles. The study of settling velocity of solid particles moving freely through a fluid contributes significantly to understanding natural processes such as sedimentation (Julien, 2010), but also in engineering, e.g. the movement of suspensions in open and closed systems (Clift, Grace, & Weber, 2005). According to Newton's second law of dynamics, a particle settling in a fluid at rest is subject to

gravity, its own buoyancy, and a resisting force, also called drag. While the first two forces do not depend on the velocity, the drag force depends on the drag coefficient and the velocity of the particle. As the particle accelerates owing to the gravity, the

fluid drags the particle until both forces are balanced and a constant or terminal velocity $w$ is achieved. Drag depends on the size, shape, density and velocity of the particle, the density and viscosity of the fluid and displays a non-linear relationship with flow hydraulics, in particular laminar or turbulent flow (Dey, Ali, & Padhi, 2019; Lapple & Sheperd, 1940). Over the years, many empirical and semi-empirical models have been proposed to compute the terminal settling velocity of natural sediments (Dietrich, 1982; Cheng, 1997; Ferguson & Church, 2004; Terfousa, Hazzabb, & Ghenaima, 2013; Goossens, 2020).

A good match of observed and predicted terminal velocity is an indication of the correct description of the dynamics of the fluid surrounding the settling particle so that the factors describing drag can be used to correctly describe fluvial and other depositional environments (Kleinhans, 2005; Lamb, Dietrich, & and Venditti, 2008). The main effort focused on the development of a unique formula able to compute the correct terminal velocity for hydraulics ranging from laminar to turbulent flow regimes around settling particles, thus reproducing the correct behaviour of the drag coefficient, $C_D$, as a function of the

Reynolds number, $Re$ describing the state of the flow. The standard $C_D$-versus- $Re$ reference curve was first obtained in 1940 by Lapple-Sheppard (Lapple & Sheperd, 1940) by fitting tabulated data from 17 different authors for spherical particles. For either the laminar or the turbulent regime, the relationship between $C_D$ and $Re$ is well characterised by Stokes' and Newton's formula (see equations 1 and 2). However, for Reynold's numbers in the intermediate region, i.e., $1 \leq Re < 1000$, the flow is in a transitional regime and neither Stokes' nor Newton's formulations predict the experimental value of the drag coefficient

correctly. In 2004, Ferguson and Church (Ferguson & Church, 2004) proposed a formula (equation 6) derived from observations to compute the terminal velocity for all grain sizes and across all flow regimes. The proposed equation includes both the effects of viscosity and of submerged specific gravity, described by drag coefficient $C_1$ and $C_2$, respectively. These parameters take values of 18 and 0.4 for smooth spheres, respectively, but can reach greater values for typical natural sands ($C_1 = 20$, $C_2 = 1.1$) as well as very angular grains ($C_1 = 24$, $C_2 = 1.2$). Unlike many other empirical formulas, the

acceleration of gravity appears in Ferguson and Church's expression, making it suitable to predict the terminal velocity of particles settling in depositional environments with gravity different from Earth, such as Mars. As an example, using the values for smooth spheres, the Ferguson and Church formula predicts a terminal velocity of 30.1 cm s$^{-1}$ for a quartz sand spherical particle of 2 mm diameter, which corresponds to a Re of 602 and a drag coefficient $C_D$ of 0.48. These data fit well the standard reference curve (Lapple & Sheperd, 1940). However, the use of models calibrated on Earth could potentially lead to an

underestimation of sedimentation velocity on Mars, because the lower gravity on Mars will reduce settling velocity and thus drag compared to Earth. The potential error is most pronounced for Reynolds numbers between 100 and 1000 where drag increases strongly because of the transition from Stokesian to Newtonian flow (Dey et al. 2019). On Earth, sand-sized particles ranging from 100 to 2000 μm in diameter experience this change in Reynolds number because of the size-induced increase of settling velocity (Figure 1). Accordingly, the error of settling velocity models calibrated for Earth can be expected to be greatest

for particles in this size range on Mars. This raises the question whether employing terrestrial models and associated values for drag coefficients to processes on Mars causes a significant error. From this, the question of how the potential error can be

measured arises. Kuhn (2014) developed and tested an experimental apparatus to measure the sedimentation velocity of sediments of different density, size and shape and performed some specific tests on board of parabolic flights with reduced gravity. These Mars Sedimentation Experiments, MarsSedEx I and II (Kuhn, 2014), showed that measuring the settling

velocity of spherical and natural particles of approximately 500 to 1000 **μm** diameter in settling tubes is possible during a parabolic flight. The results also indicated a consistent underprediction of observed terminal velocities, which is indicative of the potential error associated with the use of drag values derived on Earth. In 2016 and 2018, the Mars Sedimentation Settling Tube Photometer Experiments, MarsSedEx-STP (Kuhn et al., 2017), designed to measure sedimentation of clouds of fine particles revealed a similar effect of reduced gravity on drag for particles ranging from 100 to 500 **μm**. Both the MarsSedEx

and MarsSedEx-STP experiments illustrated that experiments on sediment settling during parabolic flights can be used as a tool for acquiring information about fluid dynamics at different gravities. However, the design of the instruments limited the acquisition of quantitative data that enabled an exact identification of the gravity-induced errors between Earth and Mars as well as an assessment of the quality of the parabolic flight environment for measuring sedimentation. The latter involves the impact of the forces caused by the unintentional movement of the plane along its longitudinal and vertical axis during a

parabola. Furthermore, the quality of the videos used in the earlier experiments limited the tracking of individual particles along the tube due to distortions and low resolution. For example, a high number of particles was required to generate sufficient contrast in the video that captured the settling. Effectively, the movement of the front of a settling particle cloud was measured and assumed to reflect settling velocity of individual particles. However, settling velocities measured in this way represent so-called hindered settling, where particles affect each other through the flows they induce in the water column (Yin & Koch,

2007; Hagemeier, Thévenin, & Richter, 2021). Furthermore, the cylindric shape of the sediment settling tubes as well as the limitations of the video cameras available at the time caused a visual distortion during the recording of the trajectories of settling particles. Finally, the small diameter (5 cm) of the settling tubes may have caused edge effects in the flow around the particles. As improved and affordable video technology became available in recent years, combined with a redesign of the settling chambers, image acquisition which would enable the tracking of individual particles appeared possible. In this paper

we present the Computational Sedimentation Modelling Calibration (CSMC) instrument, designed to improve the detection of settling pathways of individual sediment particles. The CSMC instrument flew during the Computer Sedimentation Modelling on Mars (CompSedMars I) campaign in June 2020, and data acquisition was tested in reduced and hypergravity. The purpose of the flight was to test the operation of the instrument and assess the capabilities of the improved components, particularly regarding the procedure for calculating the terminal velocity of individual spherical particles and their settling tracks. Assessing

the capability of the CSMS instrument to deliver data with limited disturbance by side effects and good tracking of path and

velocity changes is essential to for using the data not only for calculating reduced-gravity-induced changes of drag, but also to develop and test more sophisticated models for sediment settling that capture the actual flow hydraulics.

## 2 Materials and Methods

### 2.1 Design and operation

The Computational Sedimentation Modelling Calibration instrument consists of a set of six Plexiglas (polymethacrylate) square settling chambers. The chambers are 96 by 96 mm wide (inside 80 by 80 mm) and 266 (inside 250) mm high, containing 1.6 litres of water each, or 9.6 litres in total. The walls of the chambers are made of transparent plexiglass and are 8 mm thick. The upper part of the sedimentation chamber has a 14-mm central circular opening, in which a series of two PVC ball (Cepex) valves are fitted, one on top of the other. Each ball valve contains a selection of sediments that is prepared and inserted before

flight. The chamber and the connecting outlet between the chamber and the ball valve are filled with water. In this way, when the valve is opened, the particles fall directly into the water with zero initial velocity. Figure 2 shows one of the six sedimentation chambers and the two ball valves on the top of it.

   To avoid leakage in case of a structural failure of the settling chambers, the instrument is mounted inside a Zarges box modified into a watertight glove box. The structural design and measures against leakage and other failures comply with the criteria

described in the Experimental Safety Data Package (ESPD) provided by Novespace to prepare a parabolic flight. To operate the instrument during the flight, two sets of gloves and a window are fitted to the box. The window is situated in the cover of the Zarges box. The dimensions and shape of the chambers, i.e., wider, and square instead of the cylindrical ones used in previous missions (Kuhn, 2014), were chosen to reduce the visual distortion resulting from the surface curvature of tubes and edge effects caused by interaction of particles with the tubes. The glove box used to transport the experimental apparatus is

800 mm length, 600 mm width and 610 mm height, with a volume of 239 litres, and a weight of 8.9 kg. A schematic picture of the side view of the Zarges box containing the sedimentation chamber is shown in Figure 3. The chambers are fixed each in an upright position onto a mounting plate, which in turn is bolted to an angled rail that connects the chambers and Zarges box to the aircraft. Two ball valves are connected to each chamber and can hold twelve sediment samples. Once a ball valve is opened, the released particles have zero initial velocity. The particles then settle for a few centimetres in the water without

the trajectory being recorded by the camera. At the time of observation, the particles have already reached a constant terminal velocity. The fully prepared experimental apparatus weighs approximately 70 kg but can still be moved by hand aboard the aircraft that performs parabolic flights.

   The settling path of the sediment is recorded by six GoPro 8 Black cameras, one per chamber, at a frame rate of 120 Hz and an array size of 1920 X 2160 pixels. The cameras were set to linear mode to avoid the typical distortion caused by the fisheye

effect. The cameras are switched on before the start of the first reduced gravity parabola. A set of 12 Osram light sticks, two for each chamber, powered by AAA batteries, was used to illuminate the inside of the box. The light sticks were attached to the box using zip ties (Figure 4, panel a). Gravity was measured using two MSR 145 loggers (MSR145). The MSR145

accelerometer is a 3-axis sensor accelerometer type, with a measurement range of ±15 g and a measurement accuracy ±0.15g (0÷5 g, 25°C) ±0.25g (5÷10g, 25°C) ±0.45g (10÷15g, 25°C). The frequency peak is 1 kHz, and the memory capacity is over 2 million values. It operates using a lithium-polymer battery in the temperature range -20 ÷ +65 ºC and has a USB interface for data transfer. The values of gravity have been recorded using a 0.1 Hz frequency. A smartphone running an app indicating gravity (e.g., g-force meter) was used to get an indication of a stable reduced gravity at the beginning of the parabola before the release of the samples. The water temperature, relevant for its viscosity, was recorded using two ibuttons placed in a settling chamber. A top view of the experimental chambers is shown in the left part of Figure 4, while the right part of the same figure shows researchers testing the proper operation of the experimental apparatus before the flight. During the flight and according to the type of experiment planned (Table 1), the bottom valve is opened once a stable gravity has been achieved and the sediments are released into the water. The bottom valve is immediately closed again and before the next parabola, the top valve is opened so that the bottom valve is loaded with sediment again.

All the experiments of the CompSedMars I mission were performed on board an airbus A310 ZERO-G operating from Dübendorf airport in Switzerland during the 4th Swiss Parabolic Flight Campaign (June the 11th 2020) (Zurich Space Hub, 2020). During a typical parabolic flight manoeuvre, the steady horizontal flight (normal gravity, $g$) is interrupted by a steep climb ("pull up"), inducing 20s of hyper gravity (1.83 $g$). Subsequently, the aircraft follows a free trajectory which depending on the angle offers approximately 33s of Martian (0.38 $g$), 24s Lunar (0.19 $g$) or 21s zero gravity, concluded by another phase of hyper gravity before returning to a terrestrial level flight gravity again. The duration of the hyper- and reduced-gravity regime is sufficiently long to perform sediment settling experiments. In fact, the particles used in the experiments reach the terminal velocity in 0.1 s (hyper gravity), 0.2 seconds (Martian gravity) and 0.5 seconds (lunar gravity).

## 2.2 Selection of particles and settling measurements

The CompSedMars I mission focused on the acquisition of highly precise data on the trajectories of settling sediment particles. To ensure the comparison to data in the literature and from previous experiments, spherical particles with a density of silicates and a size that ensured good visibility on the videos were used. The coloured spherical glass beads were provided by Microspheres-Nanospheres, USA, (microspheres-nanospheres.com, 2020). The diameter of the microspheres ranges from 1.7 to 2.0 mm, with density ranging from 2.45 to 2.5 g cm$^{-3}$ (Figure 5). Their distinct colours ensured an easy tracking of individual particles during the video analysis. The same samples were used for measurements, both at terrestrial gravity and during the parabolic flight, ensuring combined with the colour coding that the settling velocities of the same particles were compared. The size of the particles is larger than those used by Kuhn (2014). The reason for this selection was to ensure a good visibility of individual particles on the videos captured for tracking their paths. The size of the particles places them close to a Newtonian regime on Earth where drag is constant. The error of using drag coefficients from Earth for Mars is therefore expected to be smaller than for finer sands settling in a transitional regime (Figure 1). However, since the main aim of the flight was to test the suitability of the redesigned apparatus to capture particle tracks and velocity along these tracks, priority was given to the visibility of particles recorded by a simple video system rather than the measurement of the largest possible error of drag. The

drag experienced by the selected particles was estimated using the Fergusson and Church (2004) model (equation 6) for settling tests carried out at terrestrial gravity. Since $C_1$ captures the drag related to the viscosity of the liquid it is thus independent of particle size and gravity, so that only $C_2$, describing the effect of particle size and shape, has to be calibrated. The value obtained for $C_2$ is 0.36, which is slightly below the value of 0.4 suggested by Fergusson and Church (2004) for spheres of a density of 2.65 g cm$^{-3}$. We speculate that this difference is caused by the variability of particle sizes, shapes and densities. Since the difference applies to all gravities, it has no overall effect on the results of this study. Since estimates of hydrologic and hydraulic conditions in sedimentary environments based on high-resolution imagery from Mars (e.g. Williams et al. 2013, Mangold et al. 2021, Yingst et al. 2023) are naturally done without calibrating empirical models, we followed the same approach and used the value of 0.4 for $C_2$ suggested by Ferguson and Church (2004).

During the 4th Swiss Parabolic flight, sixteen parabolas (thirteen at zero, two at Martian and one at lunar gravity) were flown. Measurements were made to observe the settling of both a single isolated particle and groups of five to 10 particles at different gravities. Some samples were mixed with potassium permanganate ($KMnO_4$) grains. This allows us to have visual, not quantitative, information about the state of the fluid.

The complete list of the experiments is presented in Table 1. The selection represents a compromise between the measurement of a wide range of particle numbers in different gravities to test the quality of the particle tracking, and the replication of measurements. Therefore, just two measurements with one isolated particle were carried out in Martian and hyper-gravity. In addition, three samples with five particles each were released at Martian gravity and two during lunar. Finally, samples of 10 particles mixed with potassium permanganate grains were released during hyper, Martian, and lunar gravity parabolas. Figure 6 shows snapshots captured from the video showing particles settling during the parabolic flight.

**2.3 Video processing**

Trajectory footage was captured from GoPro cameras using Linear Field of View mode to eliminate barrel distortion (fish-eye effect) (Figure 6). The videos of the settling trajectories recorded by the GoPro cameras were cut and analysed to generate a time series of particle locations. While watching the videos with the VLC media player (VLC, 2020) the start and end times of each settling process were extracted by using the Jump to time (Previous frame) extension. Then, the videos were cut to show just the sequence with settling particles using the software ffmpeg (ffmpeg, 2020). Subsequently, all frames of each settling sequence were extracted as single images. The resulting series of images of the settling process was loaded in ImageJ (ImageJ, 2020) to perform a manual tracking of the settling particles. The first steps within ImageJ consisted of cropping the region of interest showing the settling chamber and setting the pixel to centimetre scale based on the ruler in the background. The manual tracking plugin (ImageJ, 2020) provides the basic approach of manually marking particles, as circle, in each image and writes the key parameters: track number, image number and X-Z-positions (horizontal and vertical position, respectively) to an external file. Additionally, it calculates distances and velocities of the particle between each two records based on the pixel to centimetre ratio and frames per second. The results can be visualized as small videos. A back-calculation based on the video timestamps gave the exact date and time of each frame. The gravity logger data, which has a time frequency of 10 Hz,

are then matched to the tracking records by joining them to the image with the nearest recorded time. Tables containing time, position of the particles in, and acceleration gravity along the three axes, were exported and further processed in an Excel file. Data collected during a series of tests conducted under terrestrial gravity were used to validate the procedure. Taking the ruler in the background of the chamber as a reference, we counted the number of frames during which the particle traveled 2 cm in the middle part of the chamber, which took seven to nine frames. Dividing the space traveled (2.01 cm) by the total time of the frames multiplied by the frame rate of the GoPro (0.66 s) generates a terminal velocity of 0.3 m s[-1], which agrees well with the predicted value and with the value obtained using the above-described video analysis. Detailed information on this cross-referencing can be found in Supplementary Table 1.

## 3 Results and Discussion

### 3.1 Particle Settling

Table 1 presents the list of the experiments performed within the six sedimentation chambers. During the flight, some samples got stuck as they moved from the upper valve to the lower ball valve and one GoPro camera did not record at the correct frame rate, which limited the data compared to the list presented in Table 1.

The data for the vertical position of the particles obtained from the videos were fitted by the least squares method (Canale, 2010) by fitting the position of the particles to time by a first-order polynomial function (Supplementary Table 2-4 and Supplementary Figures 1-16). By the time the cameras began to record the fall of the particles, they had already reached terminal velocity. For this reason, the slope of the polynomial of degree provides an estimate of the terminal velocity, $w$. We used this method to obtain the terminal velocities at different gravities. Knowing the experimental terminal velocity, $w$, we can compute the particles Reynolds number using the formula:

$$Re = \frac{wD}{\nu} \ , \tag{1}$$

where $D$ is the diameter of our reference particles, and $\nu$ is the kinematic viscosity, whose value has been computed from the water temperature data acquired during the flight and is equal to $9.634 \cdot 10^{-7} \ m^2 s^{-1}$. At terminal velocity, the drag force is equal to the difference between the gravity and the buoyancy force. Also, the drag force for spherical particles is equal to:

$$C_D = \rho_f \frac{A}{2} w^2, \tag{2}$$

where A is the cross-sectional area of the particle, equal to $\pi R^2$ , and $\rho_f$ is the density fluid. The experimental drag coefficient can thus be computed as:

$$C_D = \frac{(\rho_p - \rho_f) g V}{\rho_f \frac{A}{2} w^2}, \tag{3}$$

where $\rho_p$ is density and $V$ is the volume of each particle. For our analysis, we set a diameter of 1.85 mm and a density of 2.5 g cm$^{-3}$. Table 2 summarizes the results for the different gravities. For each gravity, except lunar, we compare the data for one-isolated particle (Sample 1/1) and a group of five particles (Sample 1/5 to 5/5). For lunar gravity, a group of three particles (Sample 1 and Sample 1/3 - 3/3) was identified, instead of the group of ten as planned in Experiment 6. This is due to the fact, mentioned above, that some particles got stuck in the valve and therefore did not appear in the recordings. As expected, the terminal velocity decreases with gravity, while the drag coefficients increase. The values of the terminal velocities for each gravity do not show a significant deviation between the values of an isolated particle and those of the group of three or five. This confirms that our experimental approach together with the whole apparatus allows for measuring the terminal velocity of small groups of solid spheres. The mean value and the small standard deviation are a further indication of the small dispersion of the velocity values.

To make this analysis more robust, we calculated the error associated with the calculation of the terminal velocity values using the image analysis procedure described above. As already pointed out, when the particles enter the field of view of the cameras, have already reached the terminal velocity. The terminal velocity can be estimated as the average velocity, i.e., the ratio between the vertical distance travelled by the particle, $Z$, and the time, $T$. We thus define:

$$v_{ave} = \frac{z_f - z_i}{t_f - t_i} = \frac{Z}{T} \qquad (4)$$

This is the best estimate of the velocity. The uncertainty of this measure depends on the uncertainty of the position data, which is taken equal to the sensitivity of the ruler scale on the back of each sedimentation chamber, $\Delta z = 0.001\ m$, and uncertainty on the time, $\Delta t = \frac{1}{120} s$, that corresponds to the time interval between two frames taken by the GoPros. According to the error propagation theory, the uncertainty on the velocity can be computed by:

$$\Delta v_{ave} = v_{ave} * \left( \frac{\Delta z}{Z} + \frac{\Delta z}{T} \right) \qquad (5)$$

Knowing $Z$ and $T$ from the tables produced by image analysis, we can calculate the error of the velocity values. Table 3 summarizes the terminal velocity calculation. For each gravity regime and sample (first column), the time and space interval obtained by image analysis (second and third columns), the time equation of each particle obtained by the least-squares method from these data (fourth column), and the value of the terminal velocity together with the error calculated by Equation 4 and 5 are given. The comparison between the best estimate of the terminal velocity and the associated error with the value obtained by video and image analysis shows that the potential error arising from inaccuracies of observed positions and time is less than 3%. See Table 1 in the supplementary material for the full data on the measurement accuracy.

## 3.2 Observed and estimated settling velocity

Since the experimental data are in the gravity range $1.9 < g < 18$ m s$^{-2}$, observed and predicted terminal velocities can be compared. The model developed by Ferguson and Church (FC) (Ferguson & Church, 2004) is most suitable for such a comparison because it was developed to predict the terminal velocity of particles with density of quartz and nominal diameters ranging from 0.1 to 10 mm. The expression for the terminal velocity is given by:

$$w = \frac{\Delta \rho g D^2}{C_1 \nu + \sqrt{0.75 C_2 g \Delta \rho D^3}},$$ (6)

where $C_1$ and $C_2$ are two parameters that for spherical particles are equal to 18 and 0.4, respectively, and $\Delta \rho = (\rho_p - \rho_f)/\rho_f$ is the submerged specific gravity.

Similarly, we investigated the relationship between the drag coefficient as a function of gravity. From the Ferguson and Church formula, the drag coefficient can be calculated by:

$$C_D = \left( \frac{2 C_1 \nu}{\sqrt{3 \Delta \rho g D^3}} + \sqrt{C_2} \right)^2,$$ (7)

In Tables 4,5, and 6 the settling velocities, the Reynolds numbers, and the drag coefficient for all the samples and the three gravities are reported. In addition, we compute the difference, $D = v_{obs} - v_{pred}$, between the observed and predicted physical quantities and the relative percentage difference, $D_\% = \frac{D}{v_{obs}}$.

The observed and predicted terminal velocity corresponding to the maximum computed deviation, together with the corresponding values of $D$ and $D_\%$ are presented in Tables 6 and 7 and summarised in Figure 7. At hyper and Martian gravity, we observe an underestimation of the experimental terminal velocity and Reynolds number due to the higher value of the drag coefficient. In fact, both the values of $D$ and $D_\%$ are positive, while these numbers are negative when the drag coefficient is considered (Table 6). The deviation between observed and calculated decreases with gravity and corroborate our hypothesis and earlier observations by Kuhn (2014) that the terminal velocity is underestimated when using models calibrated at terrestrial gravity for Mars. The percentage deviation value ranges from a minimum of 10.1% to a maximum of 20.3% in the case of hyper gravity sedimentation, and from 5.8% to 15.7% for Martian gravity. It is important to note that these minimum values were found for particles that were within groups of five particles. It is plausible to hypothesize that there was a slowdown, albeit small, due to particle interaction and that deviations could be even larger for single particles (Yin & Koch, 2007). The images obtained of the final sample, grains of potassium permanganate, settling in water under hyper and reduced gravity provide an indication of difference in fluid status. As can be seen from Figure 7, at hyper gravity (left side), the track appears to induce more turbulence compared to lunar gravity (right side). This is another clear indication of the different flow conditions around the particles.

Unlike hyper and Martian, at lunar gravity the predicted terminal velocity is lower than the observed one. The maximum error of the observed velocities ranges from 3.8% to 10.2%, which is lower than the deviations obtained for hyper and Martian

gravities. One possibility is that the value of this gravity is so low that flow around the particles is approaching the laminar regime where the model becomes inaccurate. However, to test this hypothesis, laminar, transitional and turbulent regimes should be explored for each gravity values by varying particle size. Such a test would also illustrate whether the observed errors in settling velocity prediction using models calibrated for Earth would affect the sorting of sand particles across a range of sizes on Mars. In turn, the analogies between terrestrial and Martian sedimentary rocks and their interpretation, e.g. with

regards to past fluvial conditions, could be assessed.

## 4 Conclusions

This study shows that the Computational Sedimentation Modelling Calibration instrument is a valid and robust experimental tool to measure the settling velocity of sediment particles at terrestrial, hyper, and reduced gravity conditions. The square sedimentation chamber and the use of the GoPro cameras with a linear field of view ensure tracking of the settling particles

without distortion. The image analysis, starting from the footage extracted by VLC software, and subsequent extraction of position as function of time by Image-J, ensure the correct computation of the terminal velocity at all gravity conditions. The error analysis shows that the error associated with the computation is small, which is also confirmed by the small values of standard deviation. The obtained data are therefore both plausible with regards to reduced gravity and drag, as well as robust with regards to potential errors when using simple, empirical models calibrated for Earth on planetary bodies with different

gravities. Improvements in the particle release mechanism will be addressed for future missions. In addition, the video system will be upgraded to capture the movement of smaller particles. The results of the experiments also confirm the results of (Kuhn, 2014) in a quantitative way and illustrate that the use of data describing fluid dynamics on Earth should be transferred to other planetary bodies with great caution. With the limitations of time and space for instruments used during parabolic flights in mind, it is also clear that such experiments must be combined with a more fundamental modelling technique which has to be

free, as far as possible, from the use of empirical or semi-empirical models, or at least their calibrated parameter values. Such a strategy would also be suitable for dealing with more complex problems where the interaction between particles becomes relevant to describe the correct flow hydraulics and sediment texture.

**Acknowledgements**

This research was conducted with the support of Swiss Space Office 2016 *Experiments for the Parabolic Flight Campaign call.* The funded proposal was entitled *Mars Sedimentation Experiment Settling Tube Photometer Rack (MarsSedEx-STP Rack).*

## Code/Data availability

The data collected for this study are available on request from the corresponding author.

## Author contribution

The settling chambers were entirely designed by Nikolaus Kuhn and built at the University of Basel. Nikolaus Kuhn and Federica Trudu performed the parabolic flight and experiments. Federica Trudu analysed the data from the GoPro recordings. Federica Trudu and Nikolaus Kuhn jointly prepared the manuscript.

## Competing interests

The contact author has declared that none of the authors has any competing interests.

**Table 1: List of experiments performed during the 4th Swiss Parabolic Flight Campaign, Dübendorf 2020. The table describes the numbered sedimentation chambers, and, for each chamber, which experiment was conducted. For example, in the first chamber, Experiment 1 is one single particle settling in hyper gravity. Additionaly, since we performed experiments with one, five or ten particles, we added Sample 1, or Sample 1-5 as a reference for Table 2.**

| Chamber 1 | Chamber 2 | Chamber 3 | Chamber 4 | Chamber 5 | Chamber 6 |
|---|---|---|---|---|---|
| Experiment 1:<br><br>One particle<br><br>Hyper gravity<br><br>Sample 1 | Experiment 3:<br><br>Five particles<br><br>Hyper gravity<br><br>Sample 1-5 | Experiment 5:<br><br>One Particle<br><br>Mars gravity<br><br>Sample 1 | Experiment 7:<br><br>Five particles<br><br>Mars gravity | Experiment 9:<br><br>Five particles<br><br>Mars gravity | Experiment 11:<br><br>Five particles<br><br>Mars gravity<br><br>Sample 1-5 |
| Experiment 2:<br><br>Ten particles mixed with $KMnO_4$<br><br>Hyper gravity | Experiment 4:<br><br>Ten particles mixed with $KMnO_4$<br><br>Mars gravity | Experiment 6:<br><br>Ten particles mixed with $KMnO_4$<br><br>Lunar gravity<br><br>Sample 1-3 | Experiment 8:<br><br>Ten particles<br><br>Mars gravity | Experiment 10:<br><br>Five particles<br><br>Lunar gravity | Experiment 12:<br><br>Five particles<br><br>Lunar gravity<br><br>Sample 1 |




**Table 2: Data for hyper, Martian, and lunar gravity. About hyper data, in Chamber 1, only a single particle in present. In Chamber 2, a group of five particles are treated as individual. Same for Martian gravity. In Chamber 3, only a single particle in present. In Chamber 6, a group of five particles are treated as individual. About lunar gravity, in Chamber 6, only a single particle in present. In Chamber 3, a group of three particles are treated as individual. In the last row after data set, mean values and standard deviations are present. Information per individual sample can be found in the Supplementary Table 5.**

| Experiment (Hyper) | Gravity (m s$^{-2}$) | w (cm s$^{-1}$) | Re | $C_d$ |
|---|---|---|---|---|
| Experiment 1/Sample 1 | 16.2 | 39.8 | 764.3 | 0.38 |
| Mean of samples of Experiment 3/Samples 1-5 | 17.2 | 43.9 | 843 | 0.34 |
| Standard deviation of samples of Experiment 3/Samples 1-5 | 0.16 | 1.6 | 29.9 | 0.02 |
| Experiment (Martian) | Gravity (m s$^{-2}$) | w (cm s$^{-1}$) | Re | $C_d$ |
| Experiment 5/Sample 1 | 3.80 | 17.2 | 330.3 | 0.48 |
| Mean of samples of Experiment 11/Samples 1-5 | 3.96 | 18.2 | 349.1 | 0.45 |
| Standard deviation of samples of Experiment 11/Samples 1-5 | 0.04 | 0.8 | 14.6 | 0.04 |
| Experiment (lunar) | Gravity (m s$^{-2}$) | w (cm s$^{-1}$) | Re | $C_d$ |
| Experiment 12/Sample 1 | 1.91 | 10.4 | 199.7 | 0.66 |
| Mean of samples of Experiment 6/Samples 1-3 | 1.91 | 9.97 | 191.3 | 0.71 |
| Standard deviation of samples of Experiment 6/Samples 1-3 | 0 | 0.21 | 3.9 | 0.03 |



**Table 3: The Table illustrates the calculation of terminal velocity by the least squares (L.S.) method from the data extracted by image analysis and the value of terminal velocity and error calculated by error propagation theory.**

| Hyper | Range of time (s) | Range of distance (cm) | $L.S.\,Equation$ | $w \pm \Delta w$ (cm·s⁻¹) |
|---|---|---|---|---|
| Sample 1/1 | 0.1 – 0.7083 | 0.206 – 24.733 | z(t) = 39.795t – 3.6981 | 40.3 ± 0.6 |
| Sample 1/5 | 0.083 – 0.675 | 0.186 – 24.628 | z(t) = 41.793t – 3.8772 | 41.3 ± 0.6 |
| Sample 2/5 | 0.233 – 0.783 | 0.165 –24.793 | z(t) = 43.707t – 9.6298 | 44.8 ± 0.7 |
| Sample 3/5 | 0.241 – 0.775 | 0.165 – 24.627 | z(t) = 44.411t – 9.7383 | 45.8 ± 0.7 |
| Sample 4/5 | 0.250 – 0.775 | 0.0413 – 24.627 | z(t) = 46.132t – 11.326 | 46.8 ± 0.7 |
| Sample 5/5 | 0.4 – 0.958 | 0.124 – 24.793 | z(t) = 43.515t – 16.92 | 44.2 ± 0.7 |
| **Martian** | **Range of time (s)** | **Range of distance (cm)** | $L.S.\,Equation$ | $w \pm \Delta w$ (cm·s⁻¹) |
| Sample 1/1 | 0.241 – 1.666 | 0.102 – 24.917 | z(t) = 17.186z – 4.5835 | 17.4 ± 0.1 |
| Sample 1/5 | 0.216 – 1.575 | 0.103 – 24.814 | z(t) = 17.794t – 3.5071 | 18.2 ± 0.1 |
| Sample 2/5 | 0.241 – 1.683 | 0.041 – 24.813 | z (t) = 17.114t – 3.9642 | 17.2 ± 0.1 |
| Sample 3/5 | 0.258 – 1.566 | 0.206 – 24.834 | z(t) = 19.071t – 4.9482 | 18.8 ± 0.1 |
| Sample 4/5 | 0.258 – 1.616 | 0.124 – 24.896 | z(t) = 18.405t – 5.0385 | 18.2 ± 0.1 |
| Sample 5/5 | 0.216 – 1.55 | 0.041 – 24.855 | z(t) = 18.468t – 4.1327 | 18.6 ± 0.1 |
| **Lunar** | **Range of time (s)** | **Range of distance (cm)** | $L.S.\,Equation$ | $w \pm \Delta w$ (cm·s⁻¹) |
| Sample 1/1 | 0.4 – 2.758 | 0.165 – 24.813 | z(t) = 10.378t – 4.1745 | 10.5 ± 0.04 |
| Sample 1/3 | 0.308 – 2.775 | 0.164 – 24.897 | z(t) = 9.8537t – 2.6857 | 10.0 ± 0.03 |
| Sample 2/3 | 0.308 – 2.825 | 0.041 – 24.979 | z(t) = 9.8385t – 3.0869 | 9.9 ± 0.03 |
| Sample 3/3 | 0.366 – 2.716 | 0.082 – 24.917 | z(t) = 10.215t – 3.0852 | 10.6 ± 0.04 |


**Table 4: Terminal velocities, terminal velocity deviation and terminal velocity percentage deviations computed for all the gravities and the samples between experimental and Ferguson and Church formula using the parameters $C_1$ and $C_2$ calibrated on Earth, $C_1$ = 18 and $C_2$ = 0.4. At hyper and Martian gravity, the settling velocities are underestimate due to the overestimate of the drag coefficient. A similar behaviour is observed for the Reynolds number.**

| Hyper observed | Hyper predicted | $D/D_\%$ | Martian observed | Martian predicted | $D/D_\%$ | Lunar observed | Lunar predicted | $D/D_\%$ |
|---|---|---|---|---|---|---|---|---|
| 39.8 | 35.8 | 4, 10.1% | 17.2 | 16.1 | 1.1, 6.4% | 10.4 | 10.8 | -0.4, -3.8% |
| 41.8 | 35.8 | 6, 14.4% | 17.8 | 16.1 | 1.7, 9.6% | 9.9 | 10.8 | -0.9, -9.1% |
| 43.7 | 35.8 | 7.9, 18.1% | 17.1 | 16.1 | 1.0, 5.8% | 9.8 | 10.8 | -1.0, -10.2% |
| 44.4 | 35.8 | 8.6, 19.4% | 19.1 | 16.1 | 3.0, 15.7% | 10.2 | 10.8 | -0.6, -5.9% |
| 46.1 | 35.8 | 10.3, 22.3% | 18.4 | 16.1 | 2.3, 12.5% | | | |
| 43.5 | 35.8 | 7.7, 17.7% | 18.5 | 16.1 | 2.4, 12.9% | | | |







**Table 5: Reynolds numbers, Reynolds number deviation and Reynolds number percentage deviation computed for all the gravities and the samples between experimental and Ferguson and Church formula using the parameters $C_1$ and $C_2$ calibrated on Earth, $C_1$ = 18 and $C_2$ = 0.4. The same trend of the Table 5 of the terminal settling velocity is observed.**

| Hyper observed | Hyper predicted | $D/D_\%$ | Martian observed | Martian predicted | $D/D_\%$ | Lunar observed | Lunar predicted | $D/D_\%$ |
|---|---|---|---|---|---|---|---|---|
| 764.3 | 687.5 | 76.9, 10.1% | 330.3 | 309.2 | 21.1, 6.4% | 199.7 | 207.4 | -7.7, -3.8% |
| 802.7 | 687.5 | 115.2, 14.4% | 341.8 | 309.2 | 33.7, 9.6% | 190.1 | 207.4 | -17.3, -9.1% |
| 839.2 | 687.5 | 151.7, 18.1% | 328.4 | 309.2 | 19.2, 5.8% | 188.2 | 207.4 | -19.2, -10.2% |
| 852.6 | 687.5 | 165.1, 19.4% | 366.8 | 309.2 | 57.6, 15.7% | 195.9 | 207.4 | -11.5, -5.9% |
| 885.3 | 687.5 | 197.8, 22.3% | 353.3 | 309.2 | 44.2, 12.5% | | | |
| 835.3 | 687.5 | 147.9, 17.7% | 355.3 | 309.2 | 46.1, 12.9% | | | |

**Table 6: Drag coefficients, drag coefficient deviation, and drag coefficient percentage deviation computed for all the gravities and the samples between experimental and Ferguson and Church formula using the parameters $C_1$ and $C_2$ calibrated on Earth, $C_1$ = 18 and $C_2$ = 0.4.**

| Hyper observed | Hyper predicted | $D/D_\%$ | Martian observed | Martian predicted | $D/D_\%$ | Lunar observed | Lunar predicted | $D/D_\%$ |
|---|---|---|---|---|---|---|---|---|
| 0.38 | 0.47 | -0.09, -23.7% | 0.48 | 0.54 | -0.06, -12.5% | 0.66 | 0.61 | 0.05, 7.6% |
| 0.36 | 0.47 | -0.11, -30.6% | 0.46 | 0.54 | -0.08, -17.4% | 0.73 | 0.61 | 0.12, 16.4% |
| 0.33 | 0.47 | -0.14, -42.4% | 0.5 | 0.54 | -0.04, -8% | 0.73 | 0.61 | 0.12, 16.4% |
| 0.32 | 0.47 | -0.15, -46.8% | 0.4 | 0.54 | -0.14, -35% | 0.68 | 0.61 | 0.07, 10.3% |
| 0.3 | 0.47 | -0.17, -56.7% | 0.43 | 0.54 | -0.11, -25.6% | | | |
| 0.34 | 0.47 | -0.13, -38.2% | 0.44 | 0.54 | -0.1, -22.7% | | | |

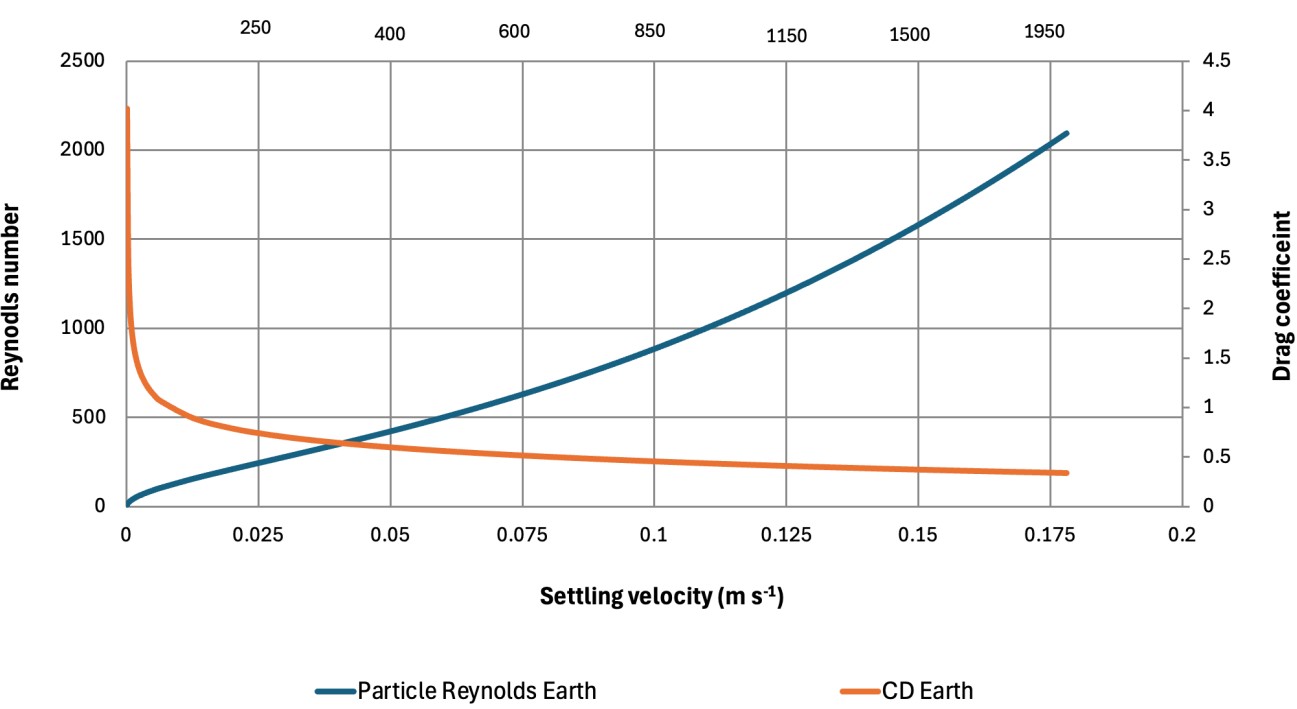


**Figure 1: Relationship between particle size and associated settling velocity, Reynolds number, drag coefficient on Earth. For fine sand up to 250 µm, drag coefficients drop very sharply, suggesting that lower gravity and associated smaller settling velocities will lead to an error when using drag coefficients from Earth on Mars.**

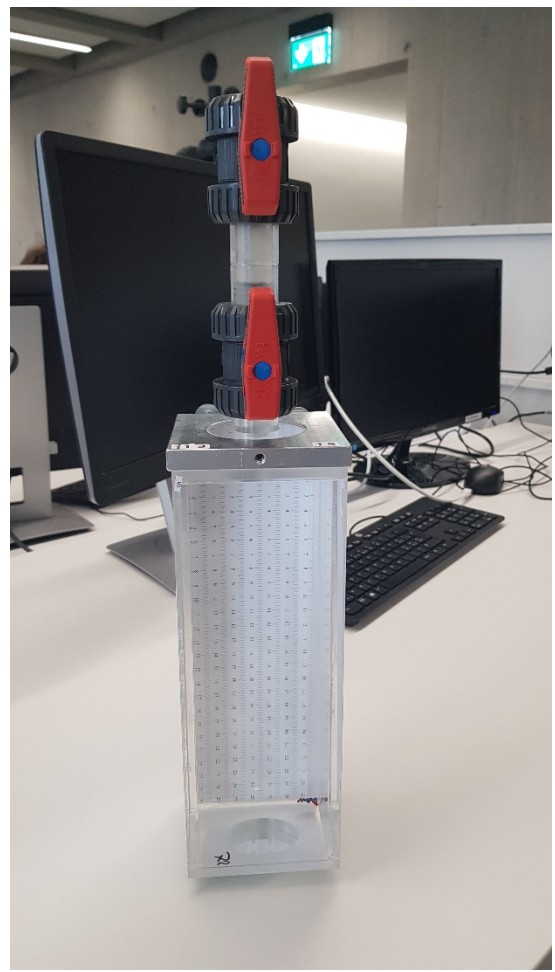

**Figure 2: One of the six sedimentation chambers with the two ball valves on top. The walls are made of transparent Plexiglas, and on the back, wall can be seen the graph paper that is used as a visual reference for trajectory analysis of the settling particles. Image credit: B. Kuhn**

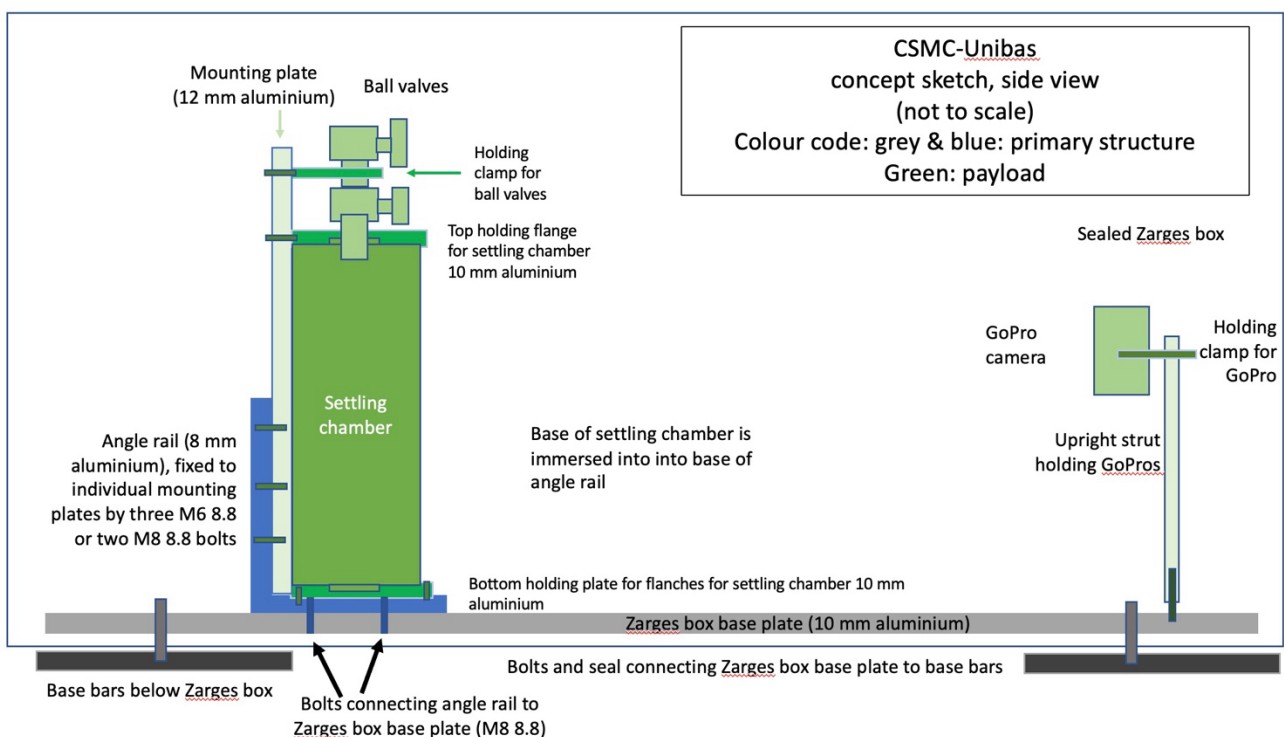

**Figure 3: Sideview of settling chamber and GoPro as fixed inside the aluminium container box. Each settling chamber is fixed to the base plate of the containing box. In front of each chamber a GoPro camera records the falling of the particles into the fluid during hyper and reduced gravity conditions.**

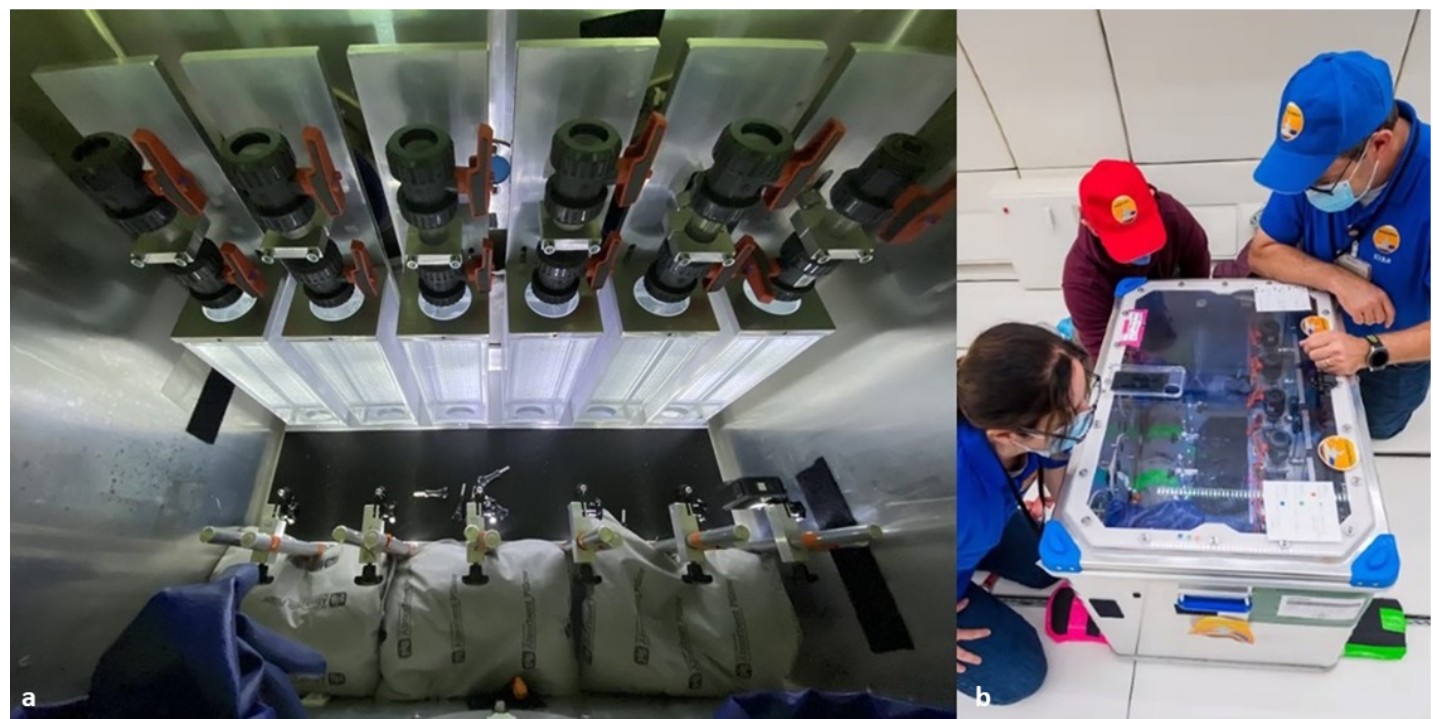

Figure 4: (a) Top view of the experimental apparatus on board the 4th Swiss Parabolic Flight Campaign held in Dübendorf Zurich (June 2020), just before the flight. The six square tubes filled with water and each mounted by two ball valves inside the aluminum containing box are shown. In front of the settling chamber are GoPro cameras and LED lighting. Behind the GoPro, absorbent pillows in case of liquid leakage, are visible. (b) The parabolic flight team as they maneuver and test the accessibility of the settling chamber using gloves inside the aircraft, the day before the parabolic flight. The team consists of four members, three of whom are visible in the figure. The blue-shirted team members, F. Trudu on the left and N. J. Kuhn on the right, flew and performed the experiments. Image credit: B. Kuhn

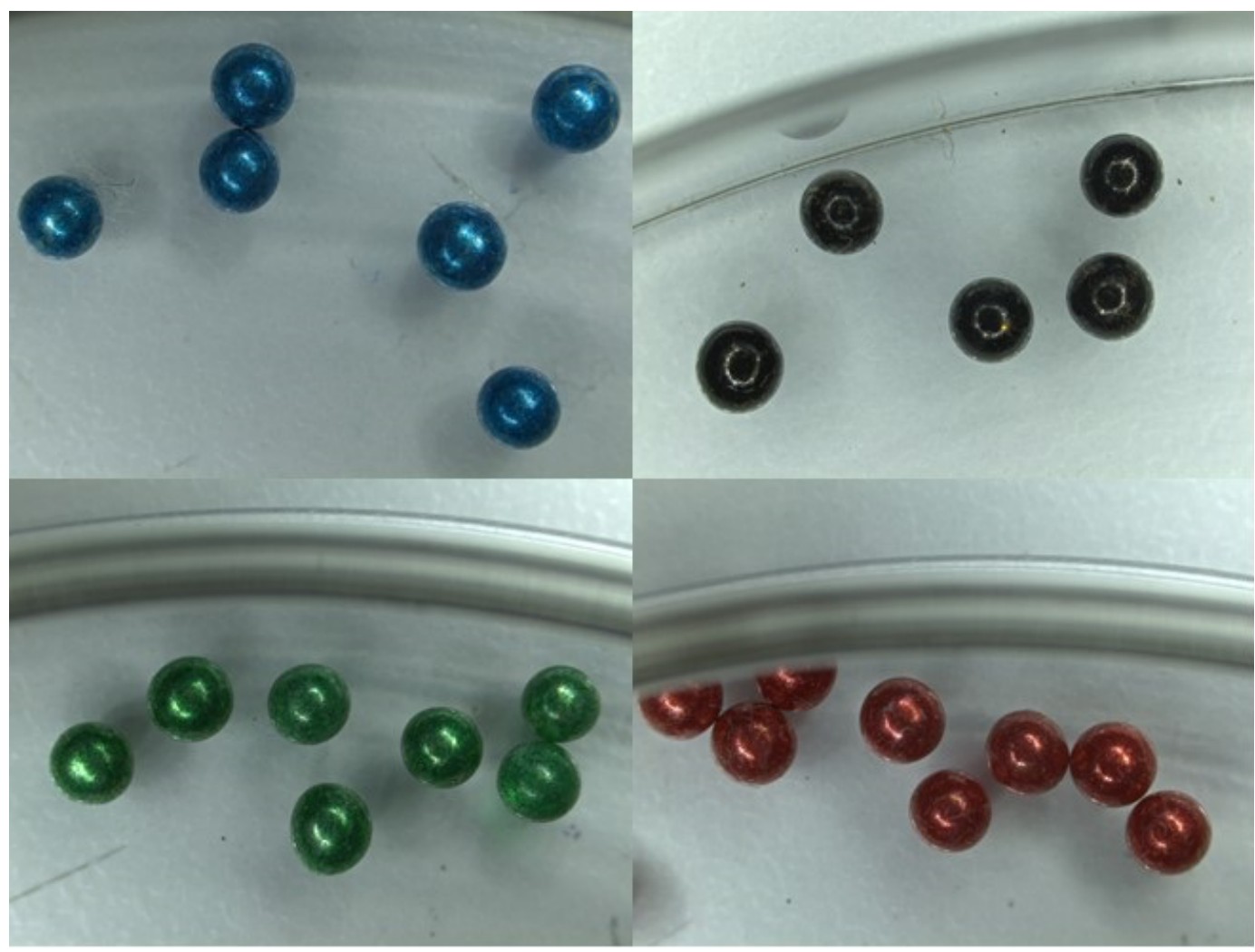

**Figure 5: Glass spheres used for CompSedMars I. The particles are all spherical, a diameter comprised between 1.7 and 2.0 mm, and have four distinct colours to be better distinguished in the particle tracking software. Image credit B. Kuhn.**

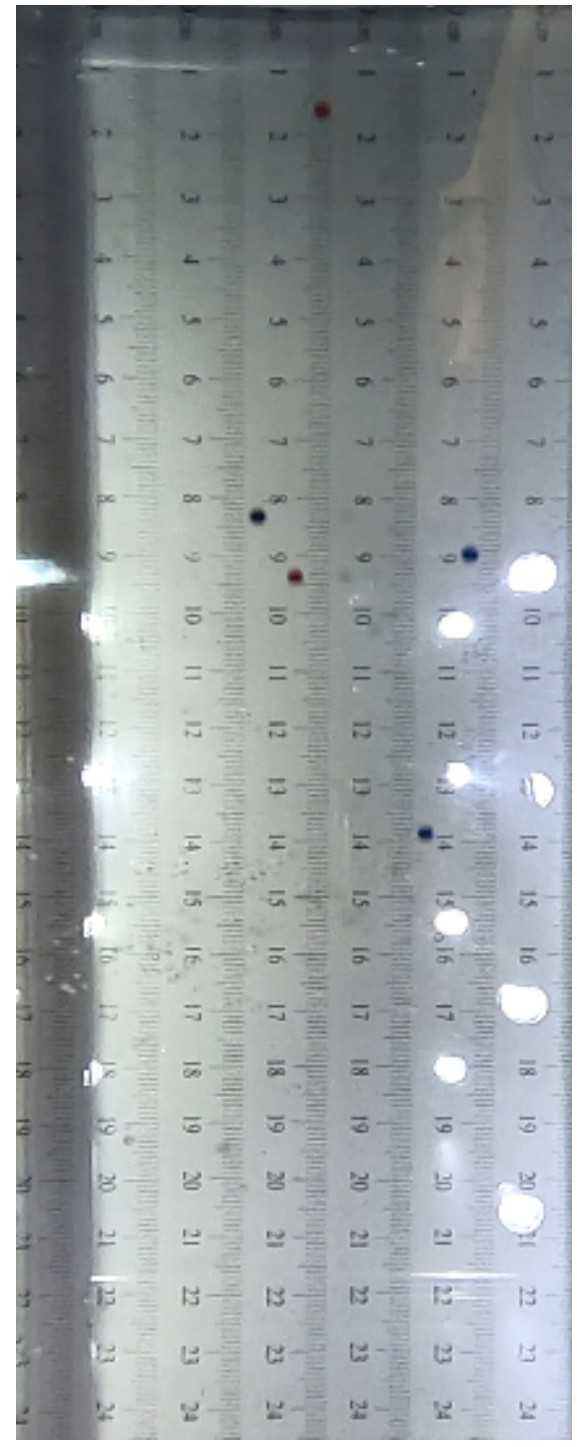


**Figure 6: Snapshot captured from a video produced by one of six Go Pros used to record the trajectories of free-falling particles in liquid water at Earth gravity conditions. Five glass reference spheres are observed. The graduated scale in the background was used as a reference to extrapolate the position of the particles as a function of time. In this picture, the lack of distortion by using the linear field of view mode of the GoPro can be seen.**


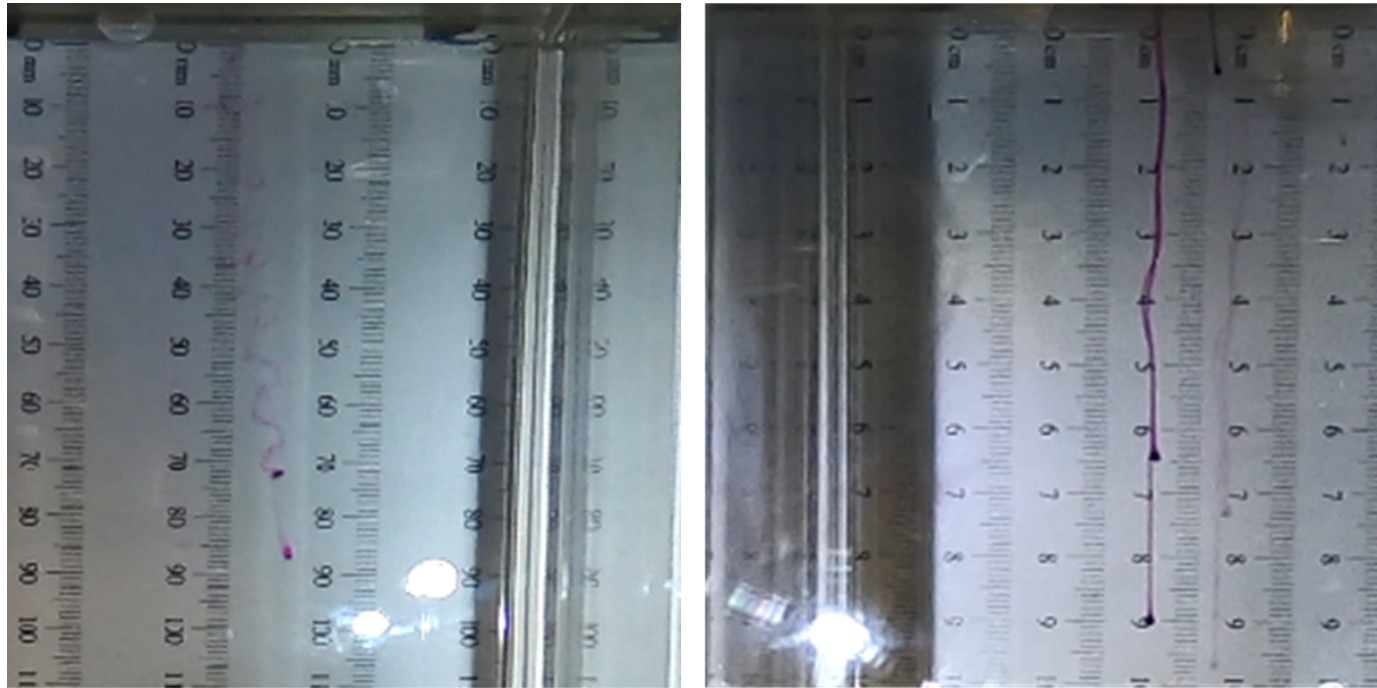

**Figure 7: Potassium permanganate grains settling at different gravities. Left panel at hyper gravity ($1.83g$), right panel at lunar gravity ($0.17g$). The change in fluid regimes from turbulent (left) to laminar (right) can be seen in the trajectories of the potassium permanganate grains. The graduated scale in mm that was placed on the background to measure the trajectories is visible in the background.**

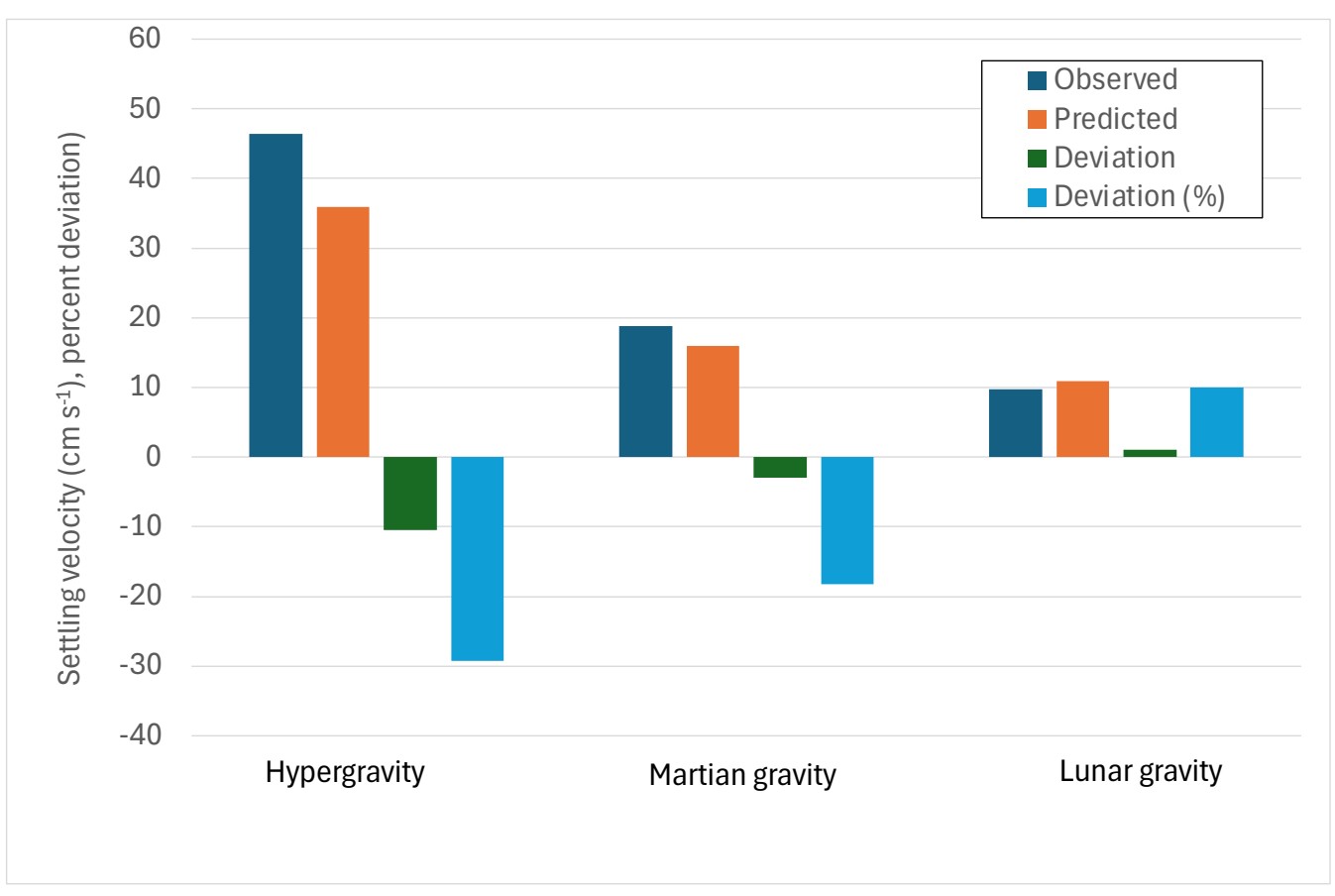


**Figure 8: Comparison of observed and predicted settling velocities based on the maximum difference between the predicted and observed terminal settling velocities at the three gravities. The deviation (=predicted – observed) is negative for Martian and hypergravity, indicating an underestimate of the predicted terminal velocities, while is slightly positive at lunar gravity, where the observed settling velocity is lower with respect to the predicted by the Ferguson and Church model.**

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
