# Peer review of "Computational Sedimentation Modelling Calibration: a tool to measure the settling velocity at different gravity conditions"

_Earth Surface Dynamics, 2023_

## Author Comment (AC1)

**Specific comments:**

**Abstract**

1. ***The first sentence of the abstract (L10) is very general. There is no explanation (here or in the introduction) why it is essential.***
   "*Research in zero or reduced gravity is essential to prepare and support planetary sciences and space exploration.*" With this sentence we want to emphasize the importance of conducting studies in environments where gravity is significantly reduced. We have added a sentence in the introduction that helps explain why these experiments are critically important:
   Conducting research in zero or reduced gravity helps simulate the conditions experienced in outer space. This is crucial for understanding how various phenomena, materials, and biological processes change under the influence of gravitational acceleration different from Earth's. For example, with reduced gravitational force, fluid flow dynamics and other physical quantities relevant to the morphology of planets and moons in our solar system**,** change. One of these physical quantities is the settling velocity of solid particles.

2. ***"models" (L20) is not defined here and therefore the sentence is unclear. There are many types of models. Please specify you mean the Ferguson and Church relation.***

3. ***"underestimate" (L21), by reading the rest of the manuscript I understand this follows from the data and previous experiments, however, without context I would think terrestrial inputs leads to overestimation of settling (because gravity might not be correctly included), not underestimation. This thought process should be explained or previous experiments should be referenced.***
   We have modified the abstract accordingly:
   Research in zero or reduced gravity is essential to prepare and support planetary sciences and space exploration. In this study, an instrument specifically designed to measure the settling velocity of sediment particles under normal, hyper-, and reduced gravity conditions is presented. The lower gravity on Mars potentially reduces drag on particles settling in water, which in turn may affect the texture of sedimentary rocks forming in a standing or moving body of water with settling particles. To assess the potential impact, an instrument was designed to simulate sediment settling at gravities different from Earth during parabolic flights. The trajectories of particles settling in water were recorded during the ascending part of a parabola (about 1.8 g), under reduced gravity conditions (Martian and lunar) and on Earth. The data were used to compute the terminal settling velocity of isolated and small groups of particles and compared to the results calculated using a semi theoretical formula derived in 2004 by Ferguson and Church (Ferguson & Church, 2004). The experimental data confirm the trend already highlighted in a series of previous experiments (Kuhn, 2014), namely that models, such as Ferguson and Church's, whose parameters were calibrated with data collected using Earth's gravity, underestimate settling velocities in reduced-gravity environments. More specifically, the values predicted using models calibrated with data collected at terrestrial gravity underestimate settling velocity on Mars. The results also demonstrate that the instrument is operational, providing a Martian gravity analogue for sedimentation studies on Earth.

**Introduction**

1. ***"Drag depends on the size, density and velocity of the particle," (L33) And shape, right? This seems to be the case according to Ferguson and Church (2004).***
   Absolutely true. We added the word 'shape' in the text.

2. ***Similar to the abstract, specify that with "models" (L56) you refer to the Ferguson and Church relation. Just "models" is too broad***.
   The term "models" is intentionally broad, because we are not referring only to the Ferguson and Church model, but to generic models that depend on a set of adjustable parameters. While it is an advantage to have so many models that allow us to calculate physical quantities, such as terminal velocity or drag

coefficients, with high accuracy and precision, the great number of adjustable parameters introduces new challenges related to the non-uniqueness of model solutions.

3. ***Similar to the abstract, I do not understand why you assume a potential for "underestimation of sedimentation velocity on Mars" (L57). I have trouble following your thought process without having read about the previous experiments or results.*** **Lower settling velocity for Mars due to lower gravity, like Ferguson and Church predict, makes sense to me. Please explain why you think it could have been an underestimation. Without context, the opposite would make more sense to me. If you calibrate everything on Earth, you might underestimate the gravity effect, so overestimate the settling velocity.**

4. ***Same issue, "underprediction" (L64), underprediction by? Feguson and Church? Compared to Earth?***

5. ***"Drag values derived on Earth" (L65). Do you mean using the same drag value for Earth and Mars does not work? Or is the predicted drag based on Mars gravity by Furguson and Church does not work?***
   I see your point. We have added a clarifying sentence in the text:
   The calculation of the terminal velocity of a solid particle in free fall in a stationary fluid depends on the force of gravity, the buoyancy force, and the drag force. While the first two forces do not depend on the velocity, the drag force depends on the drag coefficient and the velocity of the particle. The parameters of Ferguson and Church's formula were calibrated based on Earth's gravity. If the drag force has terrestrial parameters, it could be greater than it should be under reduced gravity, theoretically slowing the particle down in free fall more than under reduced gravity.

6.

**Materials and methods**

1.

2. ***Figure 2: The figure is quite clear. But I do not understand why it was not made to scale, which seems like it would be an improvement. Or would that make certain elements too small?***
   Some elements would have been too small.

3. ***Table 2: the caption of Table 2 is very unclear as it refers to Chamber number, but the reader has no information about which experiments was performed in which chamber. There is mention of a 1 particle lunar experiment and a 3-particle experiment, which are not mentioned in Table 1. The naming in the left column of the Table is in my opinion also unclear.***
   Table 1 has been modified and has the required information: chamber number, experiment, and indication for Table 2.

4. ***Table 2: I am missing the data of reference experiments on Earth. This seems valuable information and an extra data point in terms of gravity and certainly in terms of validation.***
   The same experiments conducted during the parabolic flight were conducted in Earth gravity. The data for comparison can be found in the Supplementary Information, Supplementary Table 1. We refer to these data and this Table in the text of the manuscript.

**Results and discussion**

1. **"Some samples got stuck as they moved from the upper valve to the lower ball valve" (L171). Did this not happen during your tests with Earth gravity?**
   No, it didn't happen.

2. ***Parts of the result section should be transferred to the methods section. L171-188 in section 3.1 and L214-223 section 3.2 was not measured or discovered by the authors. I am also unsure if the error determination should be in the result/discussion section. This could also be methodology or separate discussion.***
   The parts indicated serve for the discussion of the results and provide continuity to the discussion, we would prefer not to move them, as well as the determination of the error.

3. *"a group of three particles (Sample 1 and Sample 1/3 to 3/3) has been detected." (L189-190). Clarify that this was due to a problem in the experiments. Also, clarify what you mean with Sample 1, 1/3 and 3/3. These names were not defined. Is it related to number of particles, sample number or something else? Consider naming your experiments or samples 1 to 12 and indicate their planned and measured particles to avoid confusion.*

   Table 1 has been changed, as well as the description of the experiments, now numbered and an explanation of the meaning of Sample 1, etc…

4. *In my opinion it is valuable to create a graph of settling velocity (terminal fall velocity) over gravity which includes all data points of individual particles, uncertainties, and Earth experiments. In this case the reader can decide for themselves if the uncertainty is good or bad. This graph can also contain the prediction by Ferguson and Church.*

   This manuscript focuses on planetary landscapes, landforms, and their analogues. We believe that the required data would be given for another manuscript. The purpose of this manuscript is to present the experimental equipment and the results that show that this equipment, which includes video analysis, works, and can also be used by other scientists who wish to do these kinds of experiments.

5. *"Uncertainty of the position data" (L201). Despite that I think the uncertainties are reasonable, one aspect of uncertainty was not mentioned. Due to the viewing angle of the gopro and the distance between the particle and the ruler, the particle can appear in a different location. If the particle is close to the ruler the uncertainty is smaller than when the particle is closer to the gopro. The distance travelled might look larger than in reality due to the viewing angle. This could lead to overestimation of the calculated settling velocity.*

   We checked where the particles hit the bottom of the chamber. This deviation from the straight line from the release valve is small, less than a centimeter, so can be ignored. For future experiments, two cameras will be used to establish a 3d track of the particle.

6. *"It is plausible to hypothesize that there was a slowdown, due to particle interaction" (L234-235) Earlier you argue this is not the case.*

   We referred to the hindered settling phenomenon, which is plausible to be present also in such a low gravity environment condition. In this paper, we just present the instrument. More tests will allow a systematic study of the relevance of hindered settling.

7. *L237: Can you further expand on why you argue that fluid status is the cause of the difference between predicted and measured values? Why is it not and how should it be included in the Ferguson and Church relation?*

   It is explained in the Introduction. We did not want to expand the text on more basic flow hydraulics.

8. *I think it would be a good idea to also vary particle size in the future. I think it is more important information than tests with 1, 5 or 10 particles, which is in all cases likely to few particles for significant hindering to occur anyway.*

   Thank you for the suggestion, this is what we intent.

9. *Section 3.2: I am really interested to see if your tests on the ground with Earth gravity compare 100% with the predictions. If you cannot show this, it seems impossible to me to prove that the chosen values for C1 and C2 were inaccurate. These are calibration parameters. For glass spheres the difference should be minimal, but still, it is worth showing you can reproduce the predicted values with your experiments.*

   This is the point. Of course, there are some differences between our results and the predicted values, that's why a more fundamental modelling approach is required. In addition, similar standards for settling experiments should be used, e.g., identical glass spheres, e.g., with regards to roughness and coating, which affect the skin roughness of particles. The experiments only highlighted the limitation of using models that depends on empirical parameters. We are already using different and more fundamental methods, such the Lattice Boltzmann method. This latter is a computational fluid dynamics technique, that does not depend on any empirical parameters.

**Conclusions**

1. *L251-252, see previous comment, I am not convinced there is no distortion. The curvature due to the lens might be removed, but the issue of the viewing angle remains. If the particle is far away from the ruler and closer to the camera, it might appear to be at a different height then in reality. I am not sure, but I think this distortion is increased by the air-water transition.*

    We have checked the effect of the distortion by mapping settling velocities on Earth. If there was a distortion, we would see a systematic change of settling velocity along the settling path from the top of the chamber to the middle and then again back to the values observed at the top when the particle approaches the bottom. We do not observe such changes and take this as evidence for a limited, if none, effect of distortion.

**Technical corrections:**

**Abstract**

1. *"Once operational, it will be …" (L12) Is it not operational now? This sentence should not be future tense.*
2. *"… with settling particles forming a sediment" (L15) Particles are sediment. This part of the sentence should be rephrased or removed.*
3. *Remove "that" (L20)*

    Sentence has been changed according to your previous comment (Abstract, point 2, and 3.)

**Introduction**

1. *References should be merged to 1 set of brackets (L30): (Yin & Koch, 2007; Hagemeier et al., 2021). This should be corrected throughout the paper.*

    References are merged.

2. *"(see equations (1) and (2))" (L45) Brackets in brackets here are unnecessary.*

    Brackets removed.

**Materials and Methods**

1. **"is" (L108) replace by "was"**

    Done.

2. **"while density" (L136) replace by for example "with densities ranging".**

    Done.

3. **"planned measurements" (L141). Past tense, probably a remnant of a research proposal. Replace by for example "the experiments". If the planned and executed experiments are different, please specify.**

    Done.

4. **Punctuation problem at L167**

    Done.

5. **Figure 5: I only see 4 particles.**

    There are five particles, one is close to the top of the chambers, the other four are in the middle of the chamber.

**Results and Discussion**

1. *Capitalise "discussion" in section title.*

    Done.

---

## Author Response (AR2)

**Response to the reviewers: Computational Sedimentation Modelling Calibration: a tool to measure the settling velocity at different gravity conditions, By N. J. Kuhn and F. Trudu**

**Response to Referee #1**

**Dear authors,**
**Thank you for the clarifications made to the manuscript.**

**Before publication, I urge the authors to consider presenting the Earth data in the main part of the paper and make a full comparison between the Earth data, low gravity data and the Ferguson and Church model/equation, preferentially in one figure. Without this comparison, many of the statements seem unfounded. I am sure they are not, but as it is presented now, the data does not prove the statements made, e.g. "...values predicted using models calibrated with data collected at terrestrial gravity underestimate settling velocity on Mars."**

**It is necessary to show that the Earth data matches the results on the Ferguson and Church model/equation before you can state that it does not work for low gravity environments. If you do not, the reader cannot be certain your calibration parameters $C_1$ and $C_2$ of the equation are correct for the tested particles, they should be validated. A wrongly chosen $C_1$ and $C_2$ could be an explanation for the difference in results of the model/equation and the low gravity results.**

**Based on the response to the review, I understand the authors want to present this manuscript as a method paper. In my opinion the method cannot be proven without Earth data. Furthermore, the method is in my opinion not novel enough to present as only a method paper. An earlier version of this setup has already been presented in Kuhn (2014) and since then only experienced minor improvements. This setup is also not the final version, as improvements are noted for the future.**

**I am concerned that the resistance to do the comparison and showing the Earth data in the main paper is a result of salami-slicing of the data from the parabolic flight. Please prove me wrong.**

Response: We had performed a Ferguson and Church parameter optimization prior to testing our instrument and included the results now in the Methods section (section 2, lines 156 to 171). $C_1$ has to stay at 18 because it captures viscosity-induced drag, which is not affected by gravity. For terrestrial gravity, Ferguson and Church calculated a value for $C_2$ of 0.4. Fitting $C_2$ to settling velocities observed at terrestrial gravity generated a value of 0.36. The small difference illustrates the suitability of the Ferguson and Church model to simulate the settling velocities of the particles we selected for this test. We attribute the small difference of $C_2$ to small inaccuracies of particle shapes and sizes. For the further calculations in the study, we kept using the value suggested by Ferguson and Church because it reflects the nature of the error made in studies that apply non-calibrated models to sediment textures observed in high-resolution imagery from Mars. For our study, the effect of this choice is limited because of the small difference between non-calibrated and calibrated value.

We agree with the referee that the error between observation and calculation is small, but this was expected because the experiment was aimed at testing the instrument aimed at particle tracking in a parabolic flight environment. We clarified this aim in the text by describing the limitations of previous instruments in more detail (Introduction, lines 76 to 90)). We also added a figure (Figure 1) that illustrates why the effect of gravity is larger for fine than coarse sand: in the fine sand range drag values drop steeply so that the gravity (and associated settling velocity) -induced error would be much greater for a 200 µm particle than a 2 mm particle. The reason we did not use smaller particles is their limited visibility in GoPro videos. A video system that would provide sufficient resolution to capture the movement of fine sand would have to be custom-designed at costs of several ten-thousand Swiss Francs. We therefore decided to test the general suitability of parabolic flights to capture highly accurate tracks of individual settling particles first before developing an improved imaging system. Where appropriate, we clarified this aim throughout the text. We also reject the notion of salami-slicing our results, because we do not have any data on fine sands while a special issue of Earth Surface Dynamics focusing on analogue planetary environments appeared to be a good match for the scope of this study.

**Response to Referee #3**

**The role of gravity in geological processes is an important topic for a better understanding of these processes on other planets. This manuscript describes results of experiences led in parabolic flights. The results suggest a difference with models, namely an underestimation under Mars gravity. Overall, the protocol of experiences and related parameters are well described, and the results well explained, making this paper useful for the community. Nevertheless, I have a series of comments that I would like to be answered before any eventual publication.**

**One of the key results is the difference between the experiences and the model from Ferguson and Church 2004. Yet, the differences are not dramatic. For instance, Table 4 for Mars simulated gravity, the first line indicates 17.2 for the experience, and 16.1 for the model, making of this a <10% difference. This is the case for most Mars and Moon results. Yet, the error bars are not enough well explained nor plot in the key diagram of figure 7. One one hand, line 223 it is indicated that the errors on velocities are limited to less than 3%. Later, on line 253 it is written that for the moon "The maximum error of the observed velocities ranges from 3.8% to 10.2%". So why would the error be up to 10% for the experiences of the moon, but stated in general as 3% earlier?**

Response: The referee may have misunderstood the meaning of the data presented in the different parts of the text: the *"the errors on velocities are limited to less than 3%"* refers to the accuracy of the measurement listed in Table 3, which is the relative percentage between the velocity value calculated by frame count and distance covered by the particle and the value obtained by the least squares method. The 3% do therefore not refer to an error when using drag coefficient values from Earth on Mars, but the inaccuracy associated with our measurement method. The other percentage values refer to the different gravity scenarios. The way they are cited in the referee's statement are incomplete, e.g. for Mars the complete sentence reads *"The maximum error of the observed velocities ranges from 3.8% to 10.2%, which is lower than the deviations obtained for hyper and Martian gravities."*

**On the other hand, the approximation on the gravity is large. First, it is mentioned line 134 that the "angle offers approximately 33s of Martian gravity" and the table S3 indicates a variability of the g between 3.4 and 4, with a variability from flight to flight. This makes more than a difference of 10%. How was this "approximation" taken into account? Is there a diagram of g with time that could help the reader to evaluate if this approximation is of second order or not? Otherwise, it could also explain some of the difference with the model. while the results for the latter were produced for a precise Mars gravity, we can imagine that the some of the experiences may have been dominated by a gravity that was not exactly that of Mars.**

*Response:* During the parabolic flights variations in gravity occur. This was expected and our instrument was designed to cope with them. Gravity was logged at 0.1 second intervals and gravity loggers were being synchronized with the videos at one-second intervals. Consequently, for our calculations we used the actual gravity values measured during the short periods (max. 3 seconds) the particles settled through the settling chamber. The procedure has been described in the Methods section 2.3 *"The gravity logger data, which has a time frequency of 10 Hz, are then matched to the tracking records by joining them to the image with the nearest recorded time."*. The gravity data for the individual calculations are already reported in the results (Table 2).

**Again, we play here with <10% differences, and given the difficulty of measurements during these flights, this point should be well discussed. I would recommend adding error bars on the diagram of figure 7 which would make it more scientific and more convincing.**

*Response:* This issue is addressed in the reply to referee 1. To reiterate, the aim of the experiment was to test inasmuch individual particles can be tracked in a parabolic flight environment. This is a prerequisite to studies on smaller particles and studies aimed at generating data for fluid dynamics modelling. The small differences were expected because of the size of the particles.

**Also, I would be better convinced by the results if Mars and Moon gravity results would go on the same direction, and hypergravity in the other. This questions where are Earth data in this trend? Should be precisely similar to the model, right? Regard to this point, the authors mention after a previous review that they have included the data within Earth gravity (outside flights) in the Supp Table 1, but I am not sure to really understand that table, or perhaps the caption should be clarified if those are indeed under Earth gravity results. Yet, the results made with terrestrial data could be plot on the figure 7 to make clear that the experience is well set up and provide no difference with the model in that case.**

*Response:* We are not sure what the referee is addressing in this comment. The differences between observations and calculations are reported in the manuscript (Tables 4 and 5, and the new figure 8, which was formerly figure 7). The calculations are based on a model suggested and calibrated for gravity on Earth. For terrestrial gravity the model works well (see new text in section 2.2), so the only comparison that generates information with regards to the effect of gravity on drag is the difference between model and observation for the gravities that differ from Earth. However, the aim of the experiment and the manuscript is not to study this error in detail but to present parabolic flights as an environment to study processes such as sediment settling on Mars. Finally, Table 1 in the Supplementary material does not refer to an Earth-Mars comparison but reports the data on the accuracy test we performed on the method we used to calculate settling velocities from the videos. The procedure is summarized in the caption and section 3.1.

**Actually on Figure 7, the purple bars are useless and misleading. They must be removed because they are redundant from the orange bars, and they correspond to a %, not a velocity as in the Y-axis label.**

We would like to keep figure 7 (now figure 8) because it shows in a not uncommon way the absolute differences as well their percentages. The legend clarifies that the purple bar is a percentage.

**In the abstract and several other locations in the discussion, the authors mention that the model underestimate the value from the experiences, but, for the Moon, it is an overestimation. So I would suggest either use underestimate only when mentioning Mars gravity, or write it differently, for instance that the experiences provide substantial differences with the model.**
**The results at hyper gravity should be better introduced and commented. Why are they done, for which body? Any future terrestrial exoplanet? Or is this just to test the model?**

Under- and overestimation: we could not find a statement where underestimation did not just refer to Mars and/or hypergravity. The data from all the gravities were reported to give a full account of the capabilities and limitations of the instrument. Again, this manuscript has been aimed at a special issue on analogue environments with a focus on Mars. The data we have at this stage are limited and do therefore not allow much comment on the reasons for the observed and other potential differences.